# Lempel-Ziv Parsing for Sequences of Blocks

**Dmitry Kosolobov** [1],* and **Daniel Valenzuela** [2]

1    Department of Physical and Mathematical Sciences, Ural Federal University, 620000 Ekaterinburg, Russia
2    Department of Computer Science, University of Helsinki, FI-00014 Helsinki, Finland;
     daniel.valenzuela.serra@gmail.com
*    Correspondence: dkosolobov@mail.ru

**Abstract:** The Lempel-Ziv parsing (LZ77) is a widely popular construction lying at the heart of many compression algorithms. These algorithms usually treat the data as a sequence of bytes, i.e., blocks of fixed length 8. Another common option is to view the data as a sequence of bits. We investigate the following natural question: what is the relationship between the LZ77 parsings of the same data interpreted as a sequence of fixed-length blocks and as a sequence of bits (or other "elementary" letters)? In this paper, we prove that, for any integer $b > 1$, the number $z$ of phrases in the LZ77 parsing of a string of length $n$ and the number $z_b$ of phrases in the LZ77 parsing of the same string in which blocks of length $b$ are interpreted as separate letters (e.g., $b = 8$ in case of bytes) are related as $z_b = O(bz \log \frac{n}{z})$. The bound holds for both "overlapping" and "non-overlapping" versions of LZ77. Further, we establish a tight bound $z_b = O(bz)$ for the special case when each phrase in the LZ77 parsing of the string has a "phrase-aligned" earlier occurrence (an occurrence equal to the concatenation of consecutive phrases). The latter is an important particular case of parsing produced, for instance, by grammar-based compression methods.

**Keywords:** LZ77; blocks; Lempel-Ziv; SLP; grammar

## 1. Introduction

The Lempel-Ziv parsing (LZ77) [1,2] is one of the central techniques in data compression and string algorithms. Its idea is simple: to compress the data, we parse the data into phrases $f_1 f_2 \ldots f_z$ such that each phrase $f_i$ is either one letter or has an earlier occurrence in $f_1 f_2 \ldots f_i$, and the compressed encoding, instead of storing the phrases explicitly, stores the references to the occurrences. (In our investigation, we consider both non-overlapping and overlapping versions of LZ77; see precise definitions below.) Typically, algorithms that produce the parsing interpret the data as a sequence of bytes; however, some algorithms treat the data as a sequence of bits. What is the relationship between these parsings that differ only in the way they view the same input? In this paper, we investigate this question.

Our main result is that, for any integer $b > 1$, the number $z$ of phrases in the LZ77 parsing of a string of length $n$ and the number $z_b$ of phrases in the LZ77 parsing of the same string in which blocks of length $b$ are interpreted as separate letters (e.g., $b = 8$ in case of bytes) are related as $z_b = O(bz \log \frac{n}{z})$ (a more precise formulation follows). We partially complement this upper bound with a lower bound $z_b = \Omega(bz)$ in a series of examples. Further, we prove that a better bound $z_b = O(bz)$, which is tight, holds for the special case when each phrase $f_i$ in the LZ77 parsing has a "phrase-aligned" earlier occurrence, i.e., $f_i = f_j f_{j+1} \ldots f_{j'}$, for some $j < j' < i$. This special case is particularly interesting in connection to the grammar compression [3]: a grammar of size $g$ that produces the string naturally induces such "phrase-aligned" LZ77 parsing of size $O(g)$ [4].

The present work is a continuation of the line of research focused on the comparison of different efficiently computable measures of compression, mainly centering around the LZ77 parsing, the golden standard in this field (see a discussion of other close measures in [5–7]). The results on this topic are too numerous to be listed here exhaustively. We, however, point out a couple of relevant studies and open problems.

The relations between the LZ77 parsing and the grammar-based compression, which naturally induces an LZ77 parsing [4], are not sufficiently understood: known upper and lower bounds on their sizes differ by an $O(\log \log n)$ factor [8–10]. A better lower bound $z_b = \Omega(bz \log n)$, which would show that our main result is tight, even only for $b = 2$, would imply that the minimal grammar generating the string attaining this bound is of size $\Omega(z \log n)$, thus removing the $O(\log \log n)$-factor gap. This gives a new approach to attack this problem. However, it is still quite possible that our upper bound $O(bz \log \frac{n}{z})$ can be improved and lowered to $O(bz)$.

The recently introduced LZ77 variant called ReLZ [11] uses a certain preprocessing in the spirit of the so-called relative LZ77 (RLZ) [12]. The efficiency of the obtained parsing was evaluated mostly experimentally, and there are no good upper bounds comparing it to the classical LZ77. Our present result actually stems from this work in an attempt to find a good upper (or lower) bound for this version of the LZ77 parsing. We believe that techniques developed in the present paper could help to obtain such bounds.

The paper is organized as follows. The following section provides all necessary definitions and known facts about the LZ77 parsings and related concepts. In Section 3, we formalize the idea of "contracted" blocks of length $b$ and construct a series of examples showing that one of our upper bounds is tight. Then, in Section 4, we prove our upper bound results. We conclude with some open problems in Section 5.

## 2. LZ77 Parsings

A *string s* over an alphabet $\Sigma$ is a map $\{1, 2, \dots, n\} \to \Sigma$, where $n$ is referred to as the *length of s*, denoted by $|s|$. The string of length zero is called the *empty string*. We write $s[i]$ for the $i$th letter of $s$ and $s[i \dots j]$ for $s[i]s[i+1] \cdots s[j]$ (which is empty if $i > j$). A string $u$ is a *substring* of $s$ if $u = s[i \dots j]$ for some $i$ and $j$; the pair $(i, j)$ is not necessarily unique, and we say that $i$ specifies an *occurrence* of $u$ in $s$ starting at position $i$. We say that substrings $s[i \dots j]$ and $s[i' \dots j']$ *overlap* if $j \geq i'$ and $i \leq j'$. Throughout the text, substrings are sometimes identified with their particular occurrences; we do not emphasize this explicitly if it is clear from the context. We say that a substring $s[i \dots j]$ has an *earlier occurrence* if there exist $i'$ and $j'$ such that $s[i' \dots j'] = s[i \dots j]$ and $j' < i$ (note that the occurrences do not overlap). A substring $s[1 \dots j]$ (respectively, $s[i \dots n]$) is a *prefix* (respectively, *suffix*) of $s$. For any $i$ and $j$, the set $\{k \in \mathbb{Z} : i \leq k \leq j\}$ (possibly empty) is denoted by $[i \dots j]$. All logarithms have a base of two.

A parsing $s = f_1 f_2 \dots f_z$ of a given string $s$ is called a *Lempel-Ziv (LZ77) parsing* if each string $f_i$ (called *phrase*) is non-empty, and it either is one letter or occurs in the string $f_1 f_2 \dots f_{i-1}$ (i.e., it has an earlier occurrence). The *size* of the parsing is the number $z$ of phrases. The parsing is called *greedy* if it is built greedily from left to right by choosing each phrase $f_i$ as the longest substring that starts at a given position and occurs in $f_1 f_2 \dots f_{i-1}$ (see [2]). It is known that the greedy parsing is, in a sense, optimal, as it is stated in the following lemma.

**Lemma 1** (see [4,10,13]). *No LZ77 parsing of a string can have smaller size than the greedy LZ77 parsing.*

The defined LZ77 parsing is often called *non-overlapping* since its non-one-letter phrases have non-overlapping earlier occurrences. An analogously defined *overlapping LZ77 parsing* is its more popular variant: it is a parsing $f_1 f_2 \dots f_z$ in which each non-one-letter phrase $f_i$ has at least two occurrences in the string $f_1 f_2 \dots f_i$ (so that "earlier occurrences" of $f_i$ might overlap $f_i$). Consider $s = ababbabc$ for example. The greedy non-overlapping and overlapping LZ77 parsings of $s$ are $a.b.ab.ab.c$ and $a.b.abab.c$, respectively.

In this paper, we mainly discuss the non-overlapping version and, hence, for brevity, often omit the term "non-overlapping", which is assumed by default. Clearly, every LZ77 parsing is an overlapping LZ77 parsing, but the converse is not necessarily true. Indeed, one can show that the non-overlapping and overlapping LZ77 parsings are not equivalent in terms of size (see [14]). Our results, however, hold for both variants.

An LZ77 parsing $f_1 f_2 \ldots f_z$ is *phrase-aligned* if each non-one-letter phrase $f_i$ has an earlier occurrence $f_i = f_j f_{j+1} \ldots f_{j'}$, for some $j < j' < i$. This particular type of parsing is interesting because of its close connections to the grammar compression, another popular compression technique.

A *grammar* is a set of rules of the form $A \to BC$ and $A \to a$ with a designated initial rule, where $a$ denotes a letter and $A, B, C$ denote non-alphabet non-terminals; see [15]. The *size* of a grammar is the number of rules in it. A *straight line program (SLP) grammar* is a grammar that infers exactly one string. The SLP grammars and LZ77 parsings are related as follows.

**Lemma 2** (see [4,10]). *If a string s is produced by an SLP grammar of size g, then there exists a phrase-aligned LZ77 parsing $f_1 f_2 \ldots f_z$ for s of size at most g.*

By a non-constructive argument [8,10,16], one can show that the converse equivalent reduction from LZ77 parsings to SLP grammars is not possible: in some cases, the size of the minimal SLP grammar can be $\Omega(\frac{\log n}{\log \log n})$-times larger than the size of the greedy (i.e., minimal) LZ77 parsing. For completeness, let us show this by repeating here the counting argument essentially used in [10,17].

Consider all SLPs of size $g$ that produce strings over the alphabet $\{a, b\}$ and contain the rules $A \to a$ and $B \to b$. Since each rule $C \to DE$ can be constructed in, at most, $g^2$ ways by choosing a pair $(D, E)$, there are at most $g(g^2)^{g-2}$ possible configurations of the SLPs ($g - 2$ choices of the pairs $(D, E)$ and the choice of initial rule). Therefore, all such SLPs produce, at most, $g(g^2)^{g-2} \le g^{2g}$ different strings. Further, for given $n$ and $k \in [1 \ldots n-1]$, there are exactly $\binom{n}{k}$ strings of length $n$ consisting of $n - k$ letters $a$ and $k$ letters $b$, and each such string has an LZ77 parsing of size $O(k + \log n)$. If each such string can be produced by an SLP of size $g$, then $g^{2g} \ge \binom{n}{k}$ and, hence, $2g \log g \ge \log \binom{n}{k} \ge k \log \frac{n}{k}$. Choosing $k = \lceil \log n \rceil$, we deduce that $g \log g \ge \Omega(\log^2 n)$ and, therefore, $g \ge \Omega(\frac{\log^2 n}{\log \log n})$, which implies that $g \ge \Omega(k \frac{\log n}{\log \log n})$, an $\Omega(\frac{\log n}{\log \log n})$-blow up in size compared to the LZ77 parsing of size $O(k + \log n) = O(k)$.

Albeit there is no an equivalent reduction from LZ77 parsings to grammars, a slightly weaker reduction described in the following lemma still holds.

**Lemma 3** (see [4,10,18,19]). *If a string s has an (overlapping or non-overlapping) LZ77 parsing of size z, then there exists an SLP grammar of size $O(z \log \frac{n}{z})$ producing s.*

## 3. Block Contractions and a Lower Bound for Their LZ77 Parsings

Fix an integer $b > 0$. A *b-block contraction* for a string $s$ is a string $t$ such that $|t| = \lceil |s|/b \rceil$ and, for any $i, j \in [1 \ldots |t|]$, we have $t[i] = t[j]$ iff $s[b(i-1)+1 \ldots bi] = s[b(j-1)+1 \ldots bj]$. (The string $s$ is padded arbitrarily with new letters so that these two substrings are well defined.) The substrings $s[b(i-1)+1 \ldots bi]$ are called *b-blocks* or *blocks* if $b$ is clear from the context. We say that a substring $s[i \ldots j]$ *starts* (respectively, *ends*) on *a block boundary* if $i \equiv 1 \pmod{b}$ (respectively, $j \equiv 0 \pmod{b}$). A substring $s[i \ldots j]$ is *block-aligned* if it starts and ends on block boundaries.

For example, consider $s = ababbabc$ and $b = 2$. A $b$-block contraction of $s$ is $ddef$, where the letter $d$ corresponds to the blocks $ab$, $e$ corresponds to $ba$, and $f$ to $bc$. The string $ab$ has two block-aligned occurrences at positions 1 and 3 and one non-block-aligned occurrence at position 6.

In this paper, we are interested in the comparison of LZ77 parsings for a string and its $b$-block contraction. The next theorem establishes a lower bound by providing a series of examples. We will then show in the following section that this lower bound is tight for phrase-aligned LZ77 parsings.

**Theorem 1.** *For any integers $b$ and $z$ such that $1 < b \leq z/2$, there exists a string that has a phrase-aligned LZ77 parsing of size $\Theta(z)$ and whose $b$-block contraction can have only LZ77 parsings of size at least $\Omega(bz)$.*

**Proof.** Consider a string $t$ over the alphabet $\{a, b\}$ whose greedy LZ77 parsing is phrase-aligned and has size $z' = z - b$. For $h \in [0 \dots b-1]$, define a morphism $\phi_h$ such that $\phi_h(a) = c^{b-h-1}ac^h$ and $\phi_h(b) = c^{b-h-1}bc^h$, where $c$ is a letter different from $a$ and $b$. Note that $|\phi_h(a)| = |\phi_h(b)| = b$. The example string is $s = \phi_0(t)c^b\phi_1(t)\phi_2(t)\dots\phi_{b-2}(t)$. It has a phrase-aligned LZ77 parsing of size $z = O(z' + b)$ constructed as follows. The first occurrences of the substrings $\phi_0(a)$ and $\phi_0(b)$ take $O(b)$ phrases to encode; therefore, $\phi_0(t)$ can be encoded into $O(z' + b)$ phrases by mimicking the parsing of $t$. Further, each substring $\phi_h(t)$, for $h > 0$, occurs at position $h + 1$ and, thus, can be encoded by one phrase referring to this occurrence. The latter referenced occurrence is phrase-aligned provided the prefix $s[1 \dots b]$ and the substring $s[|\phi_0(t)| + 1 \dots |\phi_0(t)| + b] = c^b$ were parsed trivially as sequences of $b$ letters.

At the same time, any LZ77 parsing of a $b$-block contraction of the string $s$ must have size at least $(b-1)z'$: the $b$-block contraction of each substring $\phi_h(t)$ produces a "copy" of $t$ over a different alphabet, but the new "contracted" letters of these "copies" are pairwise distinct; therefore, each contraction should be parsed without references to other substrings, hence, occupying at least $z'$ phrases. It remains to note that $(b-1)z' = (b-1)(z-b) \geq (b-1)z/2 = \Omega(bz)$.  □

## 4. Upper Bounds on LZ77 Parsings for Block Contractions

We first consider the case of phrase-aligned LZ77 parsings in the following theorem. As will be seen later, it easily implies our result for arbitrary parsings. The proof of this theorem is quite complicated and long, occupying most of the present section.

**Theorem 2.** *Suppose that $f_1 f_2 \dots f_z$ is a phrase-aligned LZ77 parsing of a string $s$, i.e., every non-one-letter phrase $f_i$ has an earlier occurrence $f_i = f_j f_{j+1} \dots f_{j'}$, for some $j < j' < i$. Then, for any integer $b > 1$, the size of the greedy LZ77 parsing of a $b$-block contraction for $s$ is $O(bz)$.*

**Proof.** Let $s$ be a given string and $s_b$ be a $b$-block contraction of $s$. Assume that the alphabets of $s$ and $s_b$ do not intersect. We are to show that there exists *some* LZ77 parsing of $s_b$ of size $O(bz)$. Then, the statement of the theorem immediately follows from Lemma 1. The proof is split into five Sections 4.1–4.5.

### 4.1. Basic Ideas

For simplicity, assume that $|s|$ is a multiple of $b$ (if not, $s$ is padded with at most $b$ new letters). Our construction relies on the following notion: for $h \in [0..b-1]$, the *$h$-shifted block parsing of $s$* is the following parsing for the string $s[h+1..|s|-b+h]$:

$$s[h+1..|s|-b+h] = t_{h,1}d_{h,1}t_{h,2}d_{h,2}\dots t_{h,z'-1}d_{h,z'-1}t_{h,z'},$$

where $|d_{h,1}| = |d_{h,2}| = \dots = |d_{h,z'-1}| = b$, and each phrase $t_{h,i}$ is the longest substring starting at position $p = h + |t_{h,1}d_{h,1}\dots t_{h,i-1}d_{h,i-1}| + 1$, whose length is a multiple of $b$ and that is entirely contained in the phrase $f_j$ covering the position $p$, i.e., $0 < p - |f_1 f_2 \dots f_{j-1}| \leq |f_j|$ and $p + |t_{h,i}| - 1 \leq |f_1 f_2 \dots f_j|$ (in particular, $t_{h,i}$ might be empty if $f_j$ is too short). It might be viewed also as a parsing constructed by a greedy process that, starting from the position $h + 1$, alternatively chooses first a longest substring $t_{h,i}$ whose length is a multiple of $b$ that starts at the current position and is "inscribed" in the current phrase $f_j$, and then chooses a phrase $d_{h,i}$ of length exactly $b$, which "bridges" neighboring phrases in $f_1 f_2 \dots f_z$. It is straightforward that $z' \leq 2z$. It is instructive to imagine the $h$-shifted block parsings (for $h = 0, 1, \dots, b-1$) written one after another as in Figure 1. Let us describe briefly the rationale behind the definition.

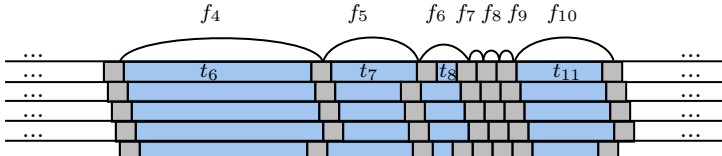

**Figure 1.** The $h$-shifted block parsings for $h = 0, 1, 2, 3, 4$. Gray and blue rectangles denote, respectively, phrases $d_{h,i}$ and $t_{h,i}$. Here, we have $b = 5$. Note that some phrases $t_{h,i}$ are empty and, thus, not depicted.

In the case $h = 0$, we omit the first index "$h$" in the notation $t_{h,i}, d_{h,i}$ and write the zero-shifted block parsing as $t_1 d_1 t_2 d_2 \ldots d_{z'-1} t_{z'}$. Note that the phrases in this parsing are block-aligned, and each phrase $t_i$ has an earlier occurrence; however, it is not an LZ77 parsing of $s$ since substrings $d_i$ might not have earlier occurrences. Moreover, the zero-shifted block parsing could have served as a generator of a correct LZ77 parsing for $s_b$ if every substring $t_i$ had a block-aligned earlier occurrence in $s$: then, the required parsing for $s_b$ would have consisted of $2z' - 1$ phrases obtained by the $b$-block contractions of $t_1, d_1, t_2, d_2, \ldots, d_{z'-1}, t_{z'}$, respectively. (Note that each string $d_i$ has length $b$ and, thus, is contracted into one letter.) Unfortunately, albeit each substring $t_k$ has earlier occurrences in $s$, it does not necessarily have an occurrence starting and ending on block boundaries. However, each earlier occurrence $s[i \ldots j] = t_k$ is block-aligned in the $h$-shifted block parsing of $s$ with $h = j \bmod b$. This observation is the primary motivation for the introduction of the $h$-shifted block parsings, and we use it in the sequel.

Informally, our idea is as follows. Consider a phrase $t_\ell$ whose earlier occurrence $s[i \ldots j] = t_\ell$ does not start and end on block boundaries. The occurrence $s[i \ldots j]$ is block-aligned according to an $h$-shifted block parsing and, since the initial parsing $f_1 f_2 \ldots f_z$ was phrase-aligned, coincides with the concatenation of some consecutive phrases $t_{h,k} d_{h,k} t_{h,k+1} d_{h,k+1} \ldots d_{h,k'-1} t_{h,k'}$ in this $h$-shifted block parsing (details are discussed below). The phrase $t_\ell$ will be disassembled in place into smaller block-aligned chunks that have earlier occurrences starting and ending on block boundaries. Thus, a block-aligned copy of the fragment $s[i \ldots j] = t_{h,k} d_{h,k} t_{h,k+1} d_{h,k+1} \ldots d_{h,k'-1} t_{h,k'}$ will appear in place of $t_\ell$ and, if a following phrase $t_{\ell'}$ refers to the same fragment $s[i \ldots j]$, we can instead refer to this reconstructed block-aligned copy without the need to disassemble into chunks again. Suppose that each phrase $t_{h,k''}$ in the decomposition of $s[i \ldots j]$ has a block-aligned occurrence before the phrase $t_\ell$. Then, the number of chunks in place of $t_\ell$ is $O(k' - k)$: the "reconstruction" proceeds as consecutive "concatenations" of $t_{h,k}, d_{h,k}, t_{h,k+1}, d_{h,k+1}, \ldots, d_{h,k'-1}, t_{h,k'}$, skipping "concatenations" that already occurred before. There are, at most, $O(z') = O(z)$ such concatenations in total. After this, all substrings block-aligned according to the $h$-shifted block parsing will have occurrences starting and ending on block boundaries according to the string $s$ itself. For all $h \in [0..b)$, we obtain $O(bz)$ "concatenations" in total. (This informal argument requires further clarifications and details that follow below.) We note, however, that this is only an intuition, and there are many details on this way. For instance, if a phrase $t_{h,k''}$ had no block-aligned occurrences before the phrase $t_\ell$, we would have to "reconstruct" $t_{h,k''}$ by disassembling it analogously into chunks, thus recursively performing the same process. The counting of "concatenations" during the "reconstructions" has many details too.

### 4.2. Greedy Phrase-Splitting Procedure

Our construction of an LZ77 parsing for $s_b$ transforms the zero-shifted block parsing of $s$. We consecutively consider the phrases $t_1, d_1, t_2, d_2, \ldots, d_{z'-1}, t_{z'}$ from left to right: for each $d_i$, we emit a one-letter phrase—the $b$-block contraction of $d_i$—into the resulting parsing for $s_b$; for each $t_i$, we perform a process splitting $t_i$ into chunks whose lengths are multiples of $b$, and each of which either is of length $b$ or has a block-aligned earlier occurrence, and the $b$-block contractions of these chunks are the new phrases emitted into the resulting parsing for $s_b$. Let us describe the process splitting $t_i$.

Consider a phrase $t_i$ and suppose that the phrases $t_1, d_1, t_2, d_2, \ldots, d_{i-1}$ were already processed, and new phrases corresponding to them in the parsing of $s_b$ under construction were emitted. If $|t_i| = 0$, we skip this step. If $|t_i| = b$, we simply emit a new one-letter phrase by contracting $t_i$. Suppose that $|t_i| > b$. Denote by $f_\ell$ the phrase into which $t_i$ is "inscribed", i.e., $0 \le |f_1 f_2 \ldots f_\ell| - |t_1 d_1 \ldots t_{i-1} d_{i-1} t_i| < |f_\ell|$. By the assumption of the theorem, there exists a phrase-aligned earlier occurrence of $f_\ell$, i.e., there exist $j$ and $j'$ such that $j < j' < \ell$ and $f_\ell = f_j f_{j+1} \ldots f_{j'}$. Thus, there is a copy of the phrase $t_i$ in the substring $f_j f_{j+1} \ldots f_{j'}$. While the phrase $t_i$ itself is block-aligned, its copy does not have to be block-aligned. However, the copy is necessarily block-aligned in the $h$-shifted block parsing with appropriate $h$, namely with $h = (|f_1 f_2 \ldots f_{j-1}| + |t_1 d_1 t_2 d_2 \ldots d_{i-1}| - |f_1 f_2 \ldots f_{\ell-1}|) \bmod b$; see Figure 2.

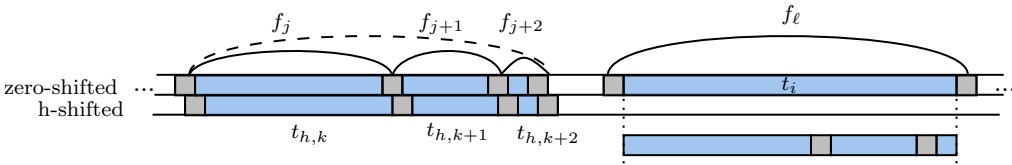

**Figure 2.** A phrase-aligned copy $f_j f_{j+1} \ldots f_{j'}$ of $f_\ell$ and the corresponding copy $t_{h,k} d_{h,k} t_{h,k+1} d_{h,k+1} \ldots d_{h,k'-1} t_{h,k'}$ of $t_i$ in the $h$-shifted block parsing. Gray and blue rectangles denote, respectively, phrases $d_{h,i}$ and $t_{h,i}$. Here, we have $j' = j + 2$ and $k' = k + 2$.

Let $t_{h,k}, d_{h,k}, t_{h,k+1}, d_{h,k+1}, \ldots, d_{h,k'-1}, t_{h,k'}$ be the phrases of the $h$-shifted block parsing that are "inscribed" into the substring $f_j f_{j+1} \ldots f_{j'}$ (the phrase $d_{h,k-1}$ could be "inscribed" too, but we omit it in this case). The key observation for our construction is that the following equality holds (see Figure 2):

$$t_i = t_{h,k} d_{h,k} t_{h,k+1} d_{h,k+1} \ldots d_{h,k'-1} t_{h,k'}.$$

Now consider the following parsing:

$$t_1 d_1 t_2 d_2 \ldots t_{i-1} d_{i-1} \cdot t_{h,k} d_{h,k} t_{h,k+1} d_{h,k+1} \ldots d_{h,k'-1} t_{h,k'}, \tag{1}$$

which is the zero-shifted block parsing of $s$ up to the phrase $t_i$ in which phrase $t_i$ was expanded into the chunks $t_{h,k}, d_{h,k}, t_{h,k+1}, d_{h,k+1}, \ldots, d_{h,k'-1}, t_{h,k'}$. If each of the chunks $t_{h,k}, t_{h,k+1}, \ldots, t_{h,k'}$ from the expansion of $t_i$ in Parsing (1) either is of length, at most, $b$ or has a block-aligned earlier occurrence, then, in principle, we could have emitted into the resulting parsing of $s_b$ the $b$-block contractions of $t_{h,k}, d_{h,k}, t_{h,k+1}, d_{h,k+1}, \ldots, d_{h,k'-1}, t_{h,k'}$, generating, at most, $2(k' - k) + 1$ new phrases for $s_b$ in total. Unfortunately, this simple approach produces too many phrases for $s_b$ in the end. Instead, we greedily unite the chunks $t_{h,k}, d_{h,k}, \ldots$ from left to right so that the resulting united chunks still either have length $b$ or have block-aligned earlier occurrences. Namely, we greedily take the maximum $m \in [k..k']$ such that the substring $t_{h,k} d_{h,k} t_{h,k+1} d_{h,k+1} \ldots d_{h,m-1} t_{h,m}$ either has length $b$ or has a block-aligned earlier occurrence (i.e., a block-aligned occurrence in the prefix $t_1 d_1 t_2 d_2 \ldots t_{i-1} d_{i-1}$); then, the $b$-block contraction of the string $q_1 = t_{h,k} d_{h,k} t_{h,k+1} d_{h,k+1} \ldots d_{h,m-1} t_{h,m}$ is emitted as a new phrase into the parsing of $s_b$ under construction. If $m = k'$, then the whole string $t_i$ was "split" into one chunk $q_1$ and we are done. Otherwise (if $m < k'$), we further emit the next (one-letter) phrase, the $b$-block contraction of $d_{h,m}$, and proceed taking the maximum $m' \in [m+1..k']$ such that the substring $q_2 = t_{h,m+1} d_{h,m+1} t_{h,m+2} d_{h,m+2} \ldots d_{h,m'-1} t_{h,m'}$ either has length $b$ or has a block-aligned earlier occurrence (i.e., a block-aligned occurrence in the prefix $t_1 d_1 t_2 d_2 \ldots t_{i-1} d_{i-1} \cdot q_1 d_{h,m}$). Similarly, we emit the contraction of $q_2$ and the one-letter contraction of $d_{h,m'}$ into the resulting parsing of $s_b$. The greedy procedure then analogously continues reading $t_i$ from left to right until the whole string $t_i$ is decomposed into chunks $q_1, d_{h,m}, q_2, d_{h,m'}, \ldots$ and all the corresponding new phrases for $s_b$ are emitted.

The described process works correctly if each of the chunks $t_{h,k}, t_{h,k+1}, \ldots, t_{h,k'}$ in the expansion of $t_i$ in Parsing (1) either has a block-aligned earlier occurrence or is of length at

most $b$. What if this is not the case? Suppose that $t_{h,k}$ is such a "bad" chunk in Parsing (1): it does not have a block-aligned earlier occurrence and $|t_{h,k}| > b$. In a certain sense, we arrive at an analogous problem that we had with the phrase $t_i$: we have to split $t_{h,k}$ (here, we are talking about the occurrence of $t_{h,k}$ from the expansion of $t_i$ in Parsing (1)) into a number of chunks whose lengths are multiples of $b$ and each of which either is of length $b$ or has a block-aligned earlier occurrence; we will then emit the $b$-block contractions of these chunks into the resulting parsing for $s_b$. We solve this problem recursively by the same procedure, which can be roughly sketched as follows. The phrase $t_{h,k}$ from the $h$-shifted block parsing of $s$ (not $t_{h,k}$ from the expansion of $t_i$ in Parsing (1)!) was "inscribed" into a phrase $f_{\ell_0}$ (such as, analogously, $t_i$ was "inscribed" into $f_\ell$), which, by the assumption of the theorem, has a phrase-aligned earlier occurrences $f_{\ell_0} = f_{j_0} f_{j_0+1} \cdots f_{j'_0}$, for some $j_0 < j'_0 < \ell_0$ (like, similarly, $f_\ell$ had an occurrence $f_\ell = f_j f_{j+1} \ldots f_{j'}$). There is, thus, a copy of $t_{h,k}$ in the substring $f_{j_0} f_{j_0+1} \ldots f_{j'_0}$. This copy is not necessarily block-aligned according to neither zero-shifted nor $h$-shifted block parsings. However, the copy is block-aligned with respect to the $h_0$-shifted block parsing for appropriate $h_0$ (for example, the copy of $t_i$ from $f_j f_{j+1} \ldots f_{j'}$ was block-aligned according to the $h$-shifted block parsing). We then consider a fragment $t_{h_0,k_0} d_{h_0,k_0} t_{h_0,k_0+1} d_{h_0,k_0+1} \ldots d_{h_0,k'_0-1} t_{h_0,k'_0}$ of the $h_0$-shifted block parsing "inscribed" into the substring $f_{j_0} f_{j_0+1} \ldots f_{j'_0}$ and conclude that this fragment is equal to $t_{h,k}$ (such as, analogously, the fragment $t_{h,k} d_{h,k} t_{h,k+1} d_{h,k+1} \ldots d_{h,k'-1} t_{h,k'}$ "inscribed" into $f_j f_{j+1} \ldots f_{j'}$ was equal to $t_i$). We proceed recursively decomposing $t_{h,k}$ into chunks by greedily uniting $t_{h_0,k_0}, d_{h_0,k_0}, t_{h_0,k_0+1}, d_{h_0,k_0+1}, \ldots, d_{h_0,k'_0-1}, t_{h_0,k'_0}$ by analogy to the procedure for $t_i$. This recursion, in turn, itself can arrive at an analogous splitting problem for another chunk (say, $t_{h_0,k'}$), which, in the same way, is solved recursively.

### 4.3. Formalized Recursive Phrase-Splitting Procedure

We formalize the described recursive procedure as a function $\mathsf{parse}(k, k', h, p)$, which is called with parameters $k, k', h, p$ such that the substring $t_{h,k} d_{h,k} t_{h,k+1} d_{h,k+1} \ldots d_{h,k'-1} t_{h,k'}$ from the $h$-shifted block parsing of $s$ occurs before the position $p$ and its copy occurs at position $p$, i.e., $h + |t_{h,1} d_{h,1} t_{h,2} d_{h,2} \ldots d_{h,k'-1} t_{h,k'}| < p$ and $s[p..p'] = t_{h,k} d_{h,k} t_{h,k+1} d_{h,k+1} \ldots d_{h,k'-1} t_{h,k'}$, for an appropriate $p' > p$. In order to process $t_i$, we call $\mathsf{parse}(k, k', h, |t_1 d_1 t_2 d_2 \ldots t_{i-1} d_{i-1}| + 1)$, where $k, k'$, and $h$ are determined as explained above. The function emits phrases of the resulting parsing of $s_b$ from left to right and works as follows:

1. If the substring $t_{h,k}$ starting at position $p$ does not have a block-aligned earlier occurrence and $|t_{h,k}| > b$, then, as was described above, we find numbers $k_0, k'_0, h_0$ such that $t_{h,k} = t_{h_0,k_0} d_{h_0,k_0} t_{h_0,k_0+1} d_{h_0,k_0+1} \ldots d_{h_0,k'_0-1} t_{h_0,k'_0}$, and we first recursively call $\mathsf{parse}(k_0, k'_0, h_0, p)$, thus processing $t_{h,k}$, then we emit a one-letter phrase by contracting $d_{h,k}$, and finally, call $\mathsf{parse}(k+1, k', h, p + |t_{h,k} d_{h,k}|)$, ending the procedure afterwards;

2. Otherwise, we compute the maximal number $m \in [k..k']$ such that the substring $q_1 = t_{h,k} d_{h,k} t_{h,k+1} d_{h,k+1} \ldots d_{h,m-1} t_{h,m}$ starting at position $p$ either has length of, at most, $b$ or has a block-aligned earlier occurrence;

3. We emit the $b$-block contraction of $q_1$ unless $q_1$ is empty (which happens if $m = k$ and $t_{h,k}$ is empty);

4. If $m = k'$, we exit; otherwise, we emit a one-letter phrase by contracting $d_{h,m}$ and call recursively $\mathsf{parse}(m+1, k', h, p + |q_1|)$.

In the sequel, we refer to the steps of the splitting procedure as "item 1, 2, 3, 4".

It is easy to see that the generated parsing of $s_b$ after processing $t_1, d_1, t_2, d_2, \ldots, d_{z'}, t_{z'}$ is a correct LZ77 parsing. It remains to estimate its size.

### 4.4. Basic Analysis of the Number of Produced Phrases

Suppose that every time a new phrase is emitted by the described procedure, we "pay" one point. We are to show that the total payment is $O(bz')$, which is $O(bz)$. For the "payment", it will suffice to reserve a budget of $7bz'$ points. In this section of the proof, we

explain how $4bz'$ points are spent (4 points per phrase $t_{h,k}$; details follow); the remaining $3bz'$ points will be discussed in the following final section of the proof.

The described process reads the string $s$ from left to right, emitting new phrases for the parsing of $s_b$ and, during the run, advances a pointer $p$, the last parameter of the function parse, from 1 to $n$. Along the way, block-aligned occurrences of some fragments from $h$-shifted block parsings appear. The key observation is that once a block-aligned occurrence of a fragment $t_{h,k}d_{h,k}t_{h,k+1}d_{h,k+1}\ldots d_{h,k'-1}t_{h,k'}$ appeared in the string $s$ at a position $p$, any subsequent block-aligned substring $t_{h,a}d_{h,a}t_{h,a+1}d_{h,a+1}\ldots d_{h,a'-1}t_{h,a'}$ with $k \le a \le a' \le k'$ that appears during the processing of phrases after the position $p$ can be united into one chunk, and there is no need to analyze it by splitting into smaller chunks again.

For each phrase $t_{h,k}$ with $k \in [1 \ldots z']$ and $h \in [0..b-1]$, we reserve four points (hence, $4bz'$ points in total). These points are spent when the string $t_{h,k}$ is first encountered in the function parse (the spending scheme is detailed below), namely when the function is invoked as $\mathrm{parse}(k_0, k_0', h_0, p)$ with parameters such that $t_{h,k} = t_{h_0,k_0}d_{h_0,k_0}t_{h_0,k_0+1}d_{h_0,k_0+1}\ldots d_{h_0,k_0'-1}t_{h_0,k_0'}$ and $t_{h,k}$ occur at the position $p$ (the position is necessarily block-aligned; therefore, this is a block-aligned occurrence of $t_{h,k}$). We note that this case includes the initial calls $\mathrm{parse}(k, k', h, p)$ that process phrases $t_i$: it is a particular case where $h = 0$. We call such an invocation of parse for the phrase $t_{h,k}$ a *generating call*. The generating call "generates" a block-aligned occurrence of $t_{h,k}$ in the string $s$. For each phrase $t_{h,k}$, at most, one generating call might happen: the only place where a generating call could have happened twice is in item 1 of the description of the function parse, but any subsequent occurrence of $t_{h,k}$ will have a block-aligned earlier occurrence, and hence, the condition in item 1 could not be satisfied. Therefore, there are, at most, $bz'$ generating calls in total, and we spend, at most, $4bz'$ points on all of them.

Let us consider the work of a generating call. For simplicity, let it be a call for a phrase $t_i$ as in the description of $\mathrm{parse}(k, k', h, p)$ above, i.e., we have $t_i = t_{h,k}d_{h,k}t_{h,k+1}d_{h,k+1}\ldots d_{h,k'-1}t_{h,k'}$, $h + |t_{h,1}d_{h,1}t_{h,2}d_{h,2}\ldots d_{h,k'-1}t_{h,k'}| < p$, and $s[p..p'] = t_i$, for an appropriate $p' > p$ (the analysis for phrases $t_{h',i}$ from other $h'$-shifted block parsings is analogous to the analysis for $t_i$). The work of parse can be essentially viewed as follows. The function parse splits $t_i$ greedily into chunks of two types:

(a)　chunks of the form $t_{h,a}d_{h,a}t_{h,a+1}d_{h,a+1}\ldots d_{h,a'-1}t_{h,a'}$ with $k \le a \le a' \le k'$ that either have a length of, at most, $b$ or have block-aligned earlier occurrences (such chunks are built in item 2 of the description of the function parse, where they are denoted as $q_1$);

(b)　chunks $t_{h,a}$ on which a generating call is invoked in item 1, which further recursively splits $t_{h,a}$ into "subchunks" (recall that, according to item 1, this is the case when $t_{h,a}$ did not have block-aligned earlier occurrences before and $|t_{h,a}| > b$).

Such possible splitting of $t_i$ is schematically depicted in Figure 3, where there is only one recursive generating call (for $t_{h,k+1}$), and there are four chunks of type (a). The $b$-block contractions of the chunks of type (a) and the contractions of short phrases $d_{h,a-1}$, $d_{h,a'}$ surrounding them are emitted as new phrases for the parsing of $s_b$. We spend the whole four-point budget allocated for $t_i$ by paying two points for the leftmost chunk of type (a) and the phrase $d_{h,a'}$ that follows after it, and by paying two points for the rightmost chunk of type (a) and the phrase $d_{h,a-1}$ that precedes it (for the example from Figure 3, these are the chunks $t_{h,k}$ and $t_{h,k+6}d_{h,k+6}t_{h,k+7}$, and the short phrases $d_{h,k}$ and $d_{h,k+5}$).

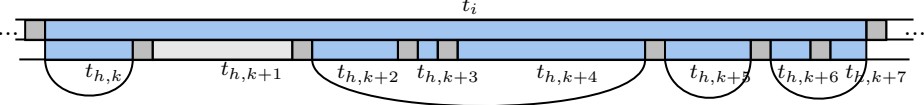

**Figure 3.** The decomposition from a generating call for $t_i = t_{h,k}d_{h,k}t_{h,k+1}d_{h,k+1}\ldots d_{h,k'-1}t_{h,k'}$. Here, $t_{h,k+1}$ is in light gray meaning that we invoke a generating call for it; for other parts, $t_i$ is split into four chunks $t_{h,k}$, $t_{h,k+2}d_{h,k+2}t_{h,k+3}d_{h,k+3}t_{h,k+4}$, $t_{h,k+5}$, $t_{h,k+6}d_{h,k+6}t_{h,k+7}$, each of which has a block-aligned earlier occurrence.

For each chunk $t_{h,a}$ of type (b), the corresponding generating call recursively produces "subchunks" splitting $t_{h,a}$. We delegate the payment for these subchunks to the budget allocated for $t_{h,a}$. It remains to pay somehow for other chunks of type (a) between the leftmost and rightmost chunks from the decomposition of $t_i$. To this end, we introduce a separate common budget of $3bz'$ points.

### 4.5. Total Number of Type (a) Chunks

We estimate the total number of chunks of type (a) by connecting them to a common combinatorial structure defined as follows. The structure is called a *segment system*: it starts with $z'$ unit integer segments $[1 \ldots 1], [2 \ldots 2], \ldots, [z' \ldots z']$, and we can perform *merge queries*, which, given two neighboring segments $[i \ldots j]$ and $[j+1 \ldots k]$ with $i \le j \le k$, create a new united segment $[i \ldots k]$ and remove the old segments $[i \ldots j]$ and $[j+1 \ldots k]$. Clearly, after $z'$ merge queries, everything will be fused into one segment $[1 \ldots z']$. Fix $h \in [1 \ldots b-1]$. We maintain a segment system connected with $h$, and we associate with each phrase $t_{h,k}$ the segment $[k \ldots k]$. During the work of the function parse, the following invariant is maintained:

- if a segment $[k..k']$ such that $k' > k$ belongs to the current segment system, then there is a block-aligned occurrence of the string $t_{h,k}d_{h,k}t_{h,k+1}d_{h,k+1} \ldots d_{h,k'-1}t_{h,k'}$ before the currently processed position $p$.

Observe that, at most, $(b-1)z' \le bz'$ merge queries can be performed in total on all $b-1$ introduced segment systems. We will spend 3 points per query from our remaining budget of $3bz'$ points. The segment systems, their merge queries, and chunks of type (a) are related as follows.

Let us temporarily alter the algorithm parse as follows: item 2 first chooses the number $m \in [k..k']$ as in the original version (as the maximal $m \in [k..k']$ such that $t_{h,k}d_{h,k}t_{h,k+1}d_{h,k+1} \ldots d_{h,m-1}t_{h,m}$ can serve as a new chunk), but then, we take the segment $[a..a']$ containing $k$ from the segment system, i.e., $a \le k \le a'$, and we assign $m := \min\{m, a'\}$. Thus, possibly, $m$ in the altered version will be smaller than in the original version. Such choice of $m$ is correct too since, due to our invariant, the altered condition still guarantees that the substring $t_{h,k}d_{h,k}t_{h,k+1}d_{h,k+1} \ldots d_{h,m-1}t_{h,m}$ is either of length $b$ or has a block-aligned earlier occurrence. It is obvious that the case when $a \ne k$ in the altered function parse can only occur for the leftmost chunk in the decomposition of $t_i$ into chunks. Further, the number of chunks of type (a) obtained for $t_i$ by the altered procedure is not less than the number of chunks of type (a) constructed by the original greedy version. It follows straightforwardly from the description that each chunk of type (a) in the altered version that is not the leftmost or rightmost chunk corresponds to a segment from the segment system associated with $h$, i.e., if $t_{h,a}d_{h,a}t_{h,a+1}d_{h,a+1} \ldots d_{h,a'-1}t_{h,a'}$, is such a chunk, then $[a \ldots a']$ belongs to the system. Similarly, the invariant implies that the unit segments $[a \ldots a]$ corresponding to chunks $t_{h,a}$ of type (b) belong to the system too. For instance, if the example of Figure 3 was obtained by this altered way, the segments $[a..k], [k+1 \ldots k+1], [k+2 \ldots k+4], [k+5 \ldots k+5], [k+6 \ldots a']$ would belong to the segment system for some $a \le k$ and $a' \ge k+7$. We then perform merge queries uniting the segments corresponding to all chunks generated for $t_i$, except the leftmost and rightmost chunks. (For the example in Figure 3, we unite all subsegments of $[k+1 \ldots k+5]$, performing two merge queries in total.) The number of merge queries for $t_i$ is equal to the number of chunks between the leftmost and rightmost chunks from the decomposition of $t_i$ minus one. The invariant of the segment system is maintained: the substring of the $h$-shifted block parsing that corresponds to the new united segment now has a block-aligned occurrence, which is precisely the occurrence in the substring $t_i$.

Suppose that we maintain in a similar way $b$ segment systems for all $h \in [0..b-1]$, performing analogous merge queries in every generating call. Then, the total number of merge queries among all $h \in [1..b-1]$ and all chunks during the work of all calls to the function parse does not exceed $bz'$ (at most, $z'$ queries for each system). Therefore, it suffices to spend the budget of $3bz'$ points for all phrases corresponding to all chunks of

type (a) as follows. We pay three points for each merge query: two for the emitted phrases corresponding to two merged chunks and one for a phrase $d_{h,a}$ separating the chunks. $\square$

As a corollary of Theorem 2, we obtain our main upper bound.

**Theorem 3.** *For any integer $b > 0$, the size $z_b$ of the greedy LZ77 parsing of a b-block contraction for a string of length n is $O(bz \log \frac{n}{z})$, where z is the size of the greedy (overlapping or non-overlapping) LZ77 parsing of this string.*

**Proof.** We apply Lemma 3 to obtain a "phrase-aligned" LZ77 parsing of size $O(z \log \frac{n}{z})$, and then, using Theorem 2, we obtain the bound $O(bz \log \frac{n}{z})$ on the size of the greedy LZ77 parsing of the $b$-block contraction. $\square$

## 5. Conclusions

Given the results obtained in this paper, the main open problem is to verify whether the upper bound from Theorem 3 is tight. To this end, one either has to improve the upper bound $z_b = O(bz \log \frac{n}{z})$ or has to provide a more elaborate series of examples improving the lower bound $z_b = \Omega(bz)$ from Section 3 (obviously, the examples must deal with non-phrase-aligned parsings). We point out, however, that the tightness of the bound from Theorem 3 would necessarily imply the tightness of the currently best upper bound $g = O(z \log \frac{n}{z})$ [4,18] from Lemma 3 that relates the size $g$ of the minimal grammar generating the string and the size $z$ of the LZ77 parsing for the string (the best lower bound up-to-date is $g = \Omega(z \frac{\log n}{\log \log n})$ [8,10]). Indeed, for a constant $b > 1$, if there exists a string whose LZ77 parsing has size $z$ and whose $b$-block contraction can have only LZ77 parsings of size at least $\Omega(z \log \frac{n}{z})$, then the minimal grammar of such string must have a size of at least $g = \Omega(z \log \frac{n}{z})$ since, by Lemma 3, the string has a phrase-aligned LZ77 parsing of size $g$, and thus, by Theorem 2, the $b$-block contraction has an LZ77 parsing of size $O(bg)$, which is $O(g)$ as $b$ is constant.

The present work stems from the paper [11] where the following LZ77 parsing was considered: we first parse the input string $s$ into phrases $p_1 p_2 \ldots p_k$ such that each phrase $p_i$ is either one letter or has an earlier occurrence in the prefix $s[1..\ell]$, for a fixed $\ell > 0$, and then we treat every phrase $p_i$ as a separate letter and construct the greedy LZ77 parsing for the obtained "contracted" string of length $k$. The resulting parsing naturally induces an LZ77 parsing for $s$. The motivation for this variant is in the considerable reduction in space during the construction of the parsing that allowed us to compress very large chunks of data using this two-pass scheme. The problem that was investigated in the present paper can be interpreted as a special case when all phrases $p_i$ have the same length $b$ (at least those phrases that are located outside of the prefix $s[1..\ell]$). The following question can be posed: can we adapt the techniques developed here to a more general case in order to find an upper bound for the LZ77 parsing from [11] in terms of the optimal greedy LZ77 parsing for $s$?

**Author Contributions:** All authors have contributed equally. All authors have read and agreed to the published version of the manuscript.

**Funding:** This research was funded by the Ministry of Science and Higher Education of the Russian Federation (Ural Mathematical Center project No. 075-02-2021-1387).

**Institutional Review Board Statement:** Not applicable.

**Informed Consent Statement:** Not applicable.

**Data Availability Statement:** Not applicable.

**Acknowledgments:** We would like to thank the anonymous referees for their comments that helped to improve the paper.

**Conflicts of Interest:** The authors declare no conflict of interest.

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
