# Peer review of "Lempel-Ziv Parsing for Sequences of Blocks"

_algorithms, doi:10.3390/a14120359_

Round 1
Reviewer 1 Report
The present paper studies the size-difference of an LZ77 encoding if applied (i) on the input word read as bit-string and (ii) as block-string (i.e., using b bits as new symbol). The result is that for any b>1 the number z of phrases in the bit-model and the number of phrases z_b in the block-model correlated on strings of length n as z_b = O(zb log(n/z)).
Strengths
- The paper studies a problem that, so I suppose, is more commonly considered in practical applications from a theoretical perspective. I found that very interesting.
- The result is somewhat tight and the proof non-trivial.
- The arguments are illustrated with helpful figures.
Weaknesses
- The paper seems to be in quite an early stage. My main criticism is actually the length: while (as a reviewer) I usually appreciate short papers, here was no reason to stay below 10 pages.
- The introduction is quite short, the motivation at a minimum, and related work shortly sketched.
- Section 2 is very condensed and overwhelms the reader with definitions and facts.
- Section 3 is a little loveless: No introduction to Theorem 1, no remarks after Theorem 2.
- No conclusion / pointers for further research.
- The proof of Theorem 1 (the main theorem) is a 4.5 page long wall of text. It would help the reader if these arguments would be divided into smaller pieces.
- The article has no conclusion and no pointer for further research. I think this is actually a no-go. The authors should reflect their work and classify their work in a bigger picture. Pointing out further research directions and open problems.
Conclusion
While the problem and the result are scientifically interesting, I don’t think the paper is in a state that can be accepted right a way. I recommend a major revision.
Notes to the Authors
- In line 20–22 you write in a little note that the non-overlapping version of the algorithm is considered. In general, it seems quite important to me that, if you prove some bounds on an algorithm, the algorithm should be in its most general version. Or, if the algorithm is restricted, it should be clear that this does not affect the obtained bounds. Hence, I think this should not be dealet with in this little note, but explained in more detail (perhaps in Section 2).
- I found Section 2 a bit “strange”. It was a, very condensed, mix of definitions and facts thrown at me that, somewhat surprisingly, then ended with examples for an upper bound.
- You may consider splitting this section into two.
- Personally, I prefer if cited lemmas / theorems are referenced as “Fact”.
- In the citation of a fact (for instance Lemma 2) do not use (e.,g.,) but provide just the source the reader is supposed to look at.
- Add some more explanatory context and eventually some examples for the definitions.
- Please divide the proof of Theorem 1 in smaller pieces.
- Formulate arguments as claims and prove them individually.
- Separate the code fragments from the proof (and actually place them in a code environment such as lstlisting).
- Prove invariants about this code in the form of some lemmas.
- Add a conclusion section that reflect the work and embed the result into known literature. Also point us to further research directions.
Minor Comments
- l2 and l8 (and across the document): typeset “i.e.” and “e.g.” as “i.\,e.” and “e.\,g.”, respectively.
- l16 “in the data compression” -> “in data compression”
- l27–37 I think it would fit to state the main theorem here (perhaps in an informal version).
- For the introduction in general, some \subsection or \paragraph could be used to divide the motivation from the results of the paper, the related work, and the structure of the manuscript.
- l106–128 This is irritating in Section 2 as it is some kind of result of the paper. Please mark it as such and formulate what is actually shown with these examples (for instance in a lemma).
- l129 “Upper Bounds” of what? Please use more meaningful headlines.
- l129–130 Here I would expect some introductory text that explains the upcoming theorem and, perhaps, sketches the plan for the proof.
- l225-l246 As noted earlier, I would suggest to separate this to a code fragment for which you can prove some properties.
- l281,283,284 “line x” is a bit odd, rather “item x” I guess.
- l375–370 I don’t know if this really has to be moved to an appendix. The paper is just 9 pages and this is an interesting note, why not incorporate it into the main text?
Author Response
We would like to thank the reviewer for the detailed review and suggestions that improved the paper. Please, find below detailed answers.
> The paper seems to be in quite an early stage. My main criticism is actually the length: while (as a reviewer) I usually appreciate short papers, here was no reason to stay below 10 pages.
> The introduction is quite short, the motivation at a minimum, and related work shortly sketched.
The introduction was not extended in the present version but we tried to compensate this with a detailed concluding section that was added. As for the rationale for the short intro mainly it is because we are not aware of direct references to the studied problem in the existing literature (which is quite surprising since the problem is natural and anybody working with LZ77 should have asked themselves this question at least once).
> Section 2 is very condensed and overwhelms the reader with definitions and facts.
We have split the section into two sections moving the part about block contractions in a separate section. We also added some additional explanations and a couple of examples between the definitions. The appendix, as suggested, was moved in Section 2 too.
> Section 3 is a little loveless: No introduction to Theorem 1, no remarks after Theorem 2.
We added some brief remarks, mainly a warning about the length of the proof and the structure of the section. It seems that there is nothing to say more here since everything motivational was discussed before and all stuff related to the proof is in the proof.
> No conclusion / pointers for further research. ... The article has no conclusion and no pointer for further research. I think this is actually a no-go. The authors should reflect their work and classify their work in a bigger picture. Pointing out further research directions and open problems.
A detailed conclusion with open problems was added.
> The proof of Theorem 1 (the main theorem) is a 4.5 page long wall of text. It would help the reader if these arguments would be divided into smaller pieces.
Indeed, the proof is difficult to consume and its formatting was not satisfactory. In order to (partially) solve the issue, we divided the proof into five subsections with captions: I Basic ideas, II Greedy phrase-splitting procedure, III Formalized recursive phrase-splitting procedure, IV Analysis of the number of produced phrases, V Total number of type (a) chunks. We also changed some formatting of the text in the proof and added a paragraph about the general idea of the proof (it is in the end of subsection I) plus several minor changes.
> In line 20–22 you write in a little note that the non-overlapping version of the algorithm is considered. In general, it seems quite important to me that, if you prove some bounds on an algorithm, the algorithm should be in its most general version. Or, if the algorithm is restricted, it should be clear that this does not affect the obtained bounds. Hence, I think this should not be dealet with in this little note, but explained in more detail (perhaps in Section 2).
We extended our result to overlapping LZ77 parsing too. Now we consider both these versions of the LZ77 parsing. The intro and abstract were changed accordingly. An explanation of differences between overlapping and non-overlapping LZ77 parsings was added in Section 2. Thank you for this suggestion, we should have done it before.
> I found Section 2 a bit “strange”. It was a, very condensed, mix of definitions and facts thrown at me that, somewhat surprisingly, then ended with examples for an upper bound.
> You may consider splitting this section into two.
We did it, thank you for the suggestion.
> Personally, I prefer if cited lemmas / theorems are referenced as “Fact”.
We decided to retain the term "lemma".
> In the citation of a fact (for instance Lemma 2) do not use (e.,g.,) but provide just the source the reader is supposed to look at.
Thank you for the correction, we removed "e.g." from citations.
> Add some more explanatory context and eventually some examples for the definitions.
We added more explanations and examples of LZ77 parsings and block contractions.
> Please divide the proof of Theorem 1 in smaller pieces.
> Formulate arguments as claims and prove them individually.
As was mentioned, we have split the proof into subsections, some of which prove claims formulated in their beginning. The claims, however, are quite intricately fused into the canvas of the proof and it is difficult to cut them into separate lemmas. We tried but decided to retain the old structure in the end.
> Separate the code fragments from the proof (and actually place them in a code environment such as lstlisting).
The discussed code of the function parse is not so formal and its informality cannot be simply removed. We tried to decorate it as a listing but it did not look well; the description requires many high-level ideas and references to a less formal description before the code. All this stuff cannot be simply omitted in order to make the code look more like a code. In the end, we decided to retain the formatting with the enumerate environment.
> Prove invariants about this code in the form of some lemmas.
The code does not have many invariants. The description of the invariants in separate lemmas would require to move too many background details from the proof into the statement of the lemma. We hope that the new separation of the proof into subsections somehow diminishes disadvantages of the current structure where everything is placed into one proof.
> Add a conclusion section that reflect the work and embed the result into known literature. Also point us to further research directions.
Thank you for the suggestion, we did it.
> Minor Comments
> l2 and l8 (and across the document): typeset “i.e.” and “e.g.” as “i.\,e.” and “e.\,g.”, respectively.
> l16 “in the data compression” -> “in data compression”
> l27–37 I think it would fit to state the main theorem here (perhaps in an informal version).
Thank you, we applied the suggested changes.
> For the introduction in general, some \subsection or \paragraph could be used to divide the motivation from the results of the paper, the related work, and the structure of the manuscript.
The intro is quite short and, in our opinion, the existing splitting into paragraphs naturally serves the same structuring goal.
> l106–128 This is irritating in Section 2 as it is some kind of result of the paper. Please mark it as such and formulate what is actually shown with these examples (for instance in a lemma).
Thank you. We stated the result with a series of examples as a theorem. Now it is called Theorem 1 and it is located in a newly created separate section (Section 3).
> l129 “Upper Bounds” of what? Please use more meaningful headlines.
We made the captions more detailed.
> l129–130 Here I would expect some introductory text that explains the upcoming theorem and, perhaps, sketches the plan for the proof.
We added the text. The sketch of the proof (as far as it could be done briefly) was added in the first subsection of the proof (it is mainly in the end of the subsection).
> l225-l246 As noted earlier, I would suggest to separate this to a code fragment for which you can prove some properties.
See an explanation above.
> l281,283,284 “line x” is a bit odd, rather “item x” I guess.
We changed "line" to "item".
> l375–370 I don’t know if this really has to be moved to an appendix. The paper is just 9 pages and this is an interesting note, why not incorporate it into the main text?
As was suggested, we moved the appendix to Section 2.
Reviewer 2 Report
The article deals with an important topic and provides a new insight. The proof provided seems to be correct; however, the readability of the article is very poor. A better presentation of the topic could make it accessible to additional readers of the journal.
Author Response
We thank the reviewer for the suggestion. We tried to improve the readability of the paper by improving the presentation of definitions, results, and proofs.
Reviewer 3 Report
The authors in the article ‘Lempel–Ziv Parsing for Sequences of Blocks’ study the size of blocks of Lempel-Ziv algorithms, which usually make use of a block size of 8 (size of a byte). The authors present the upper bound on the number of phrases for LZ77 parsing of blocks of size b. This is interesting since, with the proposed upper bound, we can now compare the algorithms which use 1-bit blocks versus those which use 8-bit (1 byte) blocks. The article presents two main theorems.
The article presents the subject clearly with necessary references for the readers.
However, I need some clarifications. Firstly, concerning Theorem 1, I feel that the authors have not clarified the significance of t and d presented in the proof of Theorem (Lines 137-147). The readers get some idea after Line 148 onwards, but the intuition behind this proposal is not very clear. The authors have clarified that the length of d is b.
Similarly, it is not clear, how one can conclude that for “payment”, a budget of 7bz’ will suffice.
Another minor remark, on line 66, it is not clear whether the authors allow i>j.
Author Response
We thank the reviewer for the provided suggestions. The answers for them are listed below.
> However, I need some clarifications. Firstly, concerning Theorem 1, I feel that the authors have not clarified the significance of t and d presented in the proof of Theorem (Lines 137-147). The readers get some idea after Line 148 onwards, but the intuition behind this proposal is not very clear. The authors have clarified that the length of d is b.
We tried to split the proof into sections and added a general idea behind the splitting into t and d in the end of the first (short) section, which briefly discusses the idea of the proof.
> Similarly, it is not clear, how one can conclude that for “payment”, a budget of 7bz’ will suffice.
We tried to be more careful with the distribution of the "payment". We added notes in the beginning of the last two section of the proof that deal with the budget points.
> Another minor remark, on line 66, it is not clear whether the authors allow i>j.
We added a clarification that this case is allowed and produces the empty string
Round 2
Reviewer 1 Report
The authors greatly improved the manuscript and incorporated most of the issues I have addressed. In particular:
- There is now a proper conclusion.
- The proof of the main theorem was splitter into 5 segments that make it easier to follow the line of argument.
- Section 2 was divided into two sections.
- The inaccuracy with overlapping / non-overlapping LZ77 was resolved.
I’m now in favor of accepting the paper.
Reviewer 3 Report
Firstly, I would like to thank the authors for considering my review comments.
The authors have provided an updated version, which now contains the following main changes
- Presence of examples for non-overlapping and overlapping LZ77 parsing, b-block contractions, b-blocks
- Splitting the proof into five clear steps.
- Clarifying the payment points: 7bz
- Inclusion of Theorem 1, which helps in further understanding their proposed upper bound values.
Minor remarks:
- Line 139 In this paper we are interesting in the comparison -> interested
- Line 427 Given the results obtained in this paper, the main remained open problem-> the main remaining open problem?